# Comprehensive Analysis of Rice Seedling Transcriptome during Dehydration and Rehydration

**DOI:** 10.3390/ijms24098439

**Published:** 2023-05-08

**Authors:** So Young Park, Dong-Hoon Jeong

**Affiliations:** 1Department of Life Science, Hallym University, Chuncheon 24252, Republic of Korea; soyoungp@hallym.ac.kr; 2Multidisciplinary Genome Institute, Hallym University, Chuncheon 24252, Republic of Korea

**Keywords:** rice, drought, rehydration, RNA-seq, differentially expressed gene

## Abstract

Drought is a harmful abiotic stress that threatens the growth, development, and yield of rice plants. To cope with drought stress, plants have evolved their diverse and sophisticated stress-tolerance mechanisms by regulating gene expression. Previous genome-wide studies have revealed many rice drought stress-responsive genes that are involved in various forms of metabolism, hormone biosynthesis, and signaling pathways, and transcriptional regulation. However, little is known about the regulation of drought-responsive genes during rehydration after dehydration. In this study, we examined the dynamic gene expression patterns in rice seedling shoots during dehydration and rehydration using RNA-seq analysis. To investigate the transcriptome-wide rice gene expression patterns during dehydration and rehydration, RNA-seq libraries were sequenced and analyzed to identify differentially expressed genes (DEGs). DEGs were classified into five clusters based on their gene expression patterns. The clusters included drought-responsive DEGs that were either rapidly or slowly recovered to control levels by rehydration treatment. Representative DEGs were selected and validated using qRT-PCR. In addition, we performed a detailed analysis of DEGs involved in nitrogen metabolism, phytohormone signaling, and transcriptional regulation. In this study, we revealed that drought-responsive genes were dynamically regulated during rehydration. Moreover, our data showed the potential role of nitrogen metabolism and jasmonic acid signaling during the drought stress response. The transcriptome data in this study could be a useful resource for understanding drought stress responses in rice and provide a valuable gene list for developing drought-resistant crop plants.

## 1. Introduction

Drought is an environmental stress that negatively affects plant growth and development, thereby causing a substantial decline in crop yields. Rice is extremely sensitive to drought stress during its entire life cycle [1]. With global climate change, variations in annual rainfall patterns, uneven distribution of rainfall in the rice-growing season, and insufficient rainfall in many areas contribute to drought stress in rice. Drought stress inhibits water uptake, decreases photosynthesis efficiency, and damages many metabolic processes, including membrane transport, energy production, and biosynthesis of many metabolites. Plants have evolved their drought avoidance and tolerance mechanisms. Drought avoidance can be achieved via enhanced water uptake by a deeper root system and reduced water loss by regulating stomatal closure. Osmotic adjustment and antioxidant accumulation are two major strategies for drought tolerance [2,3].

To understand the drought response mechanism, various genetic and molecular approaches have identified many rice genes regulated by drought conditions. Genome-wide transcriptomic approaches, such as microarray and RNA-seq, are widely used for profiling the differentially expressed genes (DEGs) under drought stress. It has been reported that more than 10% of rice genes are significantly up-regulated or down-regulated by drought stress [4,5,6,7]. Of the genes regulated by drought, metabolic process-related genes have been identified as major DEGs. Photosynthesis is the primary metabolic process that is the most sensitive to drought stress. This was due to the inhibition of CO_2_ uptake by drought-induced stomatal closure. Thus, many photosynthesis-related and carbon metabolism-associated genes are generally regulated by drought stress. Since carbon and nitrogen metabolism pathways are tightly coupled, the suppression of carbon assimilation by reduced photosynthetic efficiency affects the reprogramming of nitrogen metabolism in response to drought stress [8].

Abscisic acid (ABA) is the plant hormone that is increased under abiotic stress conditions such as drought [9,10]. Under drought stress, increased ABA levels induce many stress-responsive genes by activating transcription factors [11]. Other plant hormones are also involved in drought stress responses [12]. Antagonistic signaling crosstalk between ABA and other hormones, such as brassinosteroid, cytokinin, and auxin, contributes to drought stress responses [13,14,15]. Recent studies have reported that jasmonic acid (JA) signaling plays important roles not only in biotic stress responses, but also in abiotic stress responses, including drought [16]. JA levels are increased by drought stress, and Jasmonate ZIM-domain proteins (JAZ) and OsbHLH148 regulate drought-responsive genes in the JA signaling pathway [17,18].

Various transcription factors (TFs) regulate gene expression in response to drought stress. Of these, bZIP, AP2/EREBP, NAC, and MYB TFs are the most well-known genes that play a role in the drought stress response. Some of these TFs are activated or transcriptionally induced via ABA-dependent or ABA-independent signaling pathways during drought stress. Transgenic rice plants overexpressing these transcription factors exhibit increased drought-tolerant phenotypes [19]. For example, *OsbZIP23*-overexpressing rice plants exhibit ABA-sensitive and drought-tolerant phenotypes [20]. *OsbZIP23* is an ortholog of Arabidopsis *ABF/ABRE*, a major TF in the ABA-dependent signaling pathway. Of the TF AP2/EREBP family, OsDREB1A and OsDREB1B are key players in abiotic stress responses in an ABA-independent pathway. The overexpression of these genes results in enhanced tolerance to drought [21]. The overexpression of rice NAC transcription factors, including *OsNAC5*, *OsNAC6*, *OsNAC9*/*SNAC1*, and *OsNAC10*, also increases tolerance to drought [22,23,24,25]. In addition to these positive regulators, several TFs have been characterized as negative regulators of drought stress response. For instance, OsWRKY5 functions as a negative regulator of drought tolerance in rice [26]. The expression of OsWRKY5 is reduced by drought stress and ABA treatment. The overexpression of OsWRKY5 results in a drought-sensitive phenotype, whereas knockout exhibit enhanced drought tolerance. Although the potential use of the drought-repressed TFs for improving drought stress tolerance, in addition to drought-inducible TFs, exists, these genes have not been well reported.

Many studies have reported drought-responsive genes and their roles in drought stress response. However, little is known about the regulation of drought-responsive genes by rehydration after dehydration stress in rice. In this study, we aimed to identify the DEGs through RNA-seq analysis and to characterize the dynamics of gene expression patterns during dehydration and rehydration in rice seedlings. We also analyzed the dynamic expression patterns of genes involved in nitrogen metabolism, hormone signaling, and transcriptional regulation. The results of this study provide useful information for the development of drought-tolerant rice crops.

## 2. Results

### 2.1. Characterization of Rice Seedling Responses to Dehydration and Rehydration

To investigate the phenotypic and physiological responses of rice seedlings to dehydration and rehydration, 10-day-old rice seedlings were dehydrated for 8 h followed by rehydration for 1 or 3 days (Figure 1a). We first monitored the phenotypes of the plants exposed to dehydration and rehydration. The dehydration treatment caused leaf wilt and roll (Figure 1b). One day after re-watering, the rehydrated plants partially recovered. The three-day-rehydrated plants almost recovered from dehydration stress, and their leaves were more similar to the controls compared to the dehydrated plants. However, they remained smaller than those of the controls, and some leaf tips were still wilted.

We also measured the fresh weight of seedlings during dehydration and rehydration treatments. The dehydration treatment decreased the fresh weight by 72% compared to that of the control (Figure 1c). Compared with the dehydration-treated seedlings, rehydration for one day increased the fresh weight by 2.3 times. The fresh weight of the seedlings rehydrated for three days was similar to that of the seedlings rehydrated for one day. To determine the dehydration-induced damage to the photosynthetic systems, we measured chlorophyll fluorescence (Fv/Fm). Compared with the control plants, the 8 h dehydrated seedlings showed Fv/Fm values that were five times lower (Figure 1d). For the rehydrated seedlings, we measured the Fv/Fm values from damaged weak leaf tips and healthy recovered parts of leaves (Appendix A). The Fv/Fm values of the weak leaf tips from both 1 d rehydrated and 3 d rehydrated seedlings were approximately 0. In contrast, the healthy parts of leaves from the 1 d and 3 d rehydrated seedlings showed almost the same Fv/Fm values as the control plants. These results indicate that only some parts of the rehydrated seedlings recovered from dehydration stress.

### 2.2. Identification of Differentially Expressed Genes by Dehydration and Rehydration

To examine the transcriptomic response of rice seedlings to dehydration and rehydration, RNA-seq libraries were constructed and sequenced from two biological replicates of control: 8 h dehydrated and 1 d rehydrated seedling shoots. More than 10 million reads were obtained for each library (Appendix A). The sequence reads were mapped to the rice genome and the expression levels of genes were normalized as reads per kilobase of transcript per million mapped reads (RPKM) values. The biological replicates showed highly correlated gene expression profiles, as evidenced by Pearson’s correlation coefficients (R^2^) over 0.88 (Figure 2a). Using principal component analysis (PCA), we confirmed that two biological replicates of each treatment clustered together (Figure 2b). The first principal component (PC1) accounted for 44.56% of the variance, whereas the second principal component (PC2) accounted for 16.92% of the variance. PCA showed that the control, dehydrated, and rehydrated plants clustered apart. This result suggests that the transcriptome of the rehydration does not simply reflect recovery to that of the control condition and has a unique transcriptome composition.

To identify differentially expressed genes (DEGs), pairwise comparisons among the three conditions were performed using the EdgeR program. Thresholds of adjusted *p*-value < 0.01 and fold change > 2 were set to identify significant DEGs. A total of 8311 (4517 up-regulated and 3794 down-regulated), 2766 (1755 up-regulated and 1011 down-regulated), and 4888 (2614 up-regulated and 2274 down-regulated) transcripts were identified from the comparisons of control vs. dehydration, control vs. rehydration, and dehydration vs. rehydration, respectively (Figure 2c). These DEG groups were designated as Groups 1–6 (Appendix A). The number of overlapping transcripts among the DEG groups was analyzed to determine the correlation among the DEG groups. As a result, positive correlations were observed among Groups 1, 3, and 6. Positive correlations were also found among Groups 2, 4, and 5. However, Groups 1, 3, and 6 were negatively correlated with Groups 2, 4, and 5 (Figure 2d). This result indicates that drought-up-regulated genes (Group 1) overlapped more with rehydration-up-regulated genes (Group 3), and the genes in these two groups also overlapped with the rehydration-down-regulated genes in a comparison between dehydration and rehydration (Group 6). We also noticed that drought-down-regulated genes (Group 2) overlapped more with rehydration-down-regulated genes (Group 4), and the genes in these two groups also overlapped with the rehydration-up-regulated genes in a comparison between dehydration and rehydration (Group 5). Although our results indicated a strong correlation among the DEG groups, we also observed a notable proportion of DEGs that are unique to each DEG group. This suggests that the rehydration does not merely signify a return to the control state, and some DEGs identified through comparison with rehydration may either sustain expression levels at the dehydration state or be preferentially regulated during rehydration.

Gene ontology (GO) term enrichment was analyzed to characterize the biological functions of the genes in each DEG group (Appendix A). The most significant GO terms from each DEG group are listed in Figure 2e, and the overlapping GO terms were analyzed. We found a strong correlation between significantly enriched GO terms in the DEG groups. Genes in Groups 1, 3, and 6 were enriched with GO terms, including carbohydrate metabolism, xylan catabolic process, cell wall macromolecule catabolic process, response to stimulus, and response to abscisic acid. However, these GO terms were less represented in Groups 2, 4, and 5. Genes in Groups 2, 4, and 5 were enriched with GO terms, such as photosynthesis, nitrogen metabolism, amino acid metabolism, glutathione metabolism, and phosphorylation. This correlation was consistent with the correlation between overlapping genes among the DEGs.

### 2.3. Clustering Analysis of the Differentially Expressed Genes during Dehydration and Rehydration

To examine the gene expression patterns during dehydration and rehydration, we performed clustering analysis using the normalized RPKM values of the DEGs. A heatmap was generated based on the z-score-transformed expression levels. To reflect the major expression trends and patterns, DEGs were assigned to five major clusters using the K-mean algorithm (Figure 3a). The genes in cluster 1 were up-regulated by dehydration and their expression levels were restored by rehydration treatment. Meanwhile, the drought-up-regulated genes in cluster 2 maintained high expression levels after one day of rehydration. In contrast, the genes in clusters 3 and 4 were down-regulated by dehydration treatment. While the expression levels of cluster 3 genes were restored by rehydration, those of cluster 4 genes were still down-regulated under one-day rehydration condition. Cluster 5 genes occupied a relatively small portion compared to the other clusters and showed rehydration-preferential expression patterns.

To gain insight into the biological functions of the genes in each cluster, GO term enrichment was analyzed (Appendix A). Genes in clusters 1 and 2 were enriched in GO terms, including carbohydrate metabolism, transcriptional regulation, and nitrogen metabolic regulation. The genes involved in the water stress response were enriched only in cluster 1. Photosynthesis-related genes were highly enriched in cluster 3, whereas genes related to protein amino acid phosphorylation were enriched in cluster 4. In addition, both clusters 3 and 4 had enriched gene sets related to oxidation reduction. The genes in cluster 5, which were rehydration-preferentially expressed genes, were enriched with GO terms, including oxidative stress response and chitin metabolism. We also analyzed the Kyoto Encyclopedia of Genes and Genomes (KEGG) pathway enrichment of the gene clusters (Appendix A). We found that metabolic pathway-related genes were enriched in all clusters. The pathways involved in carbon metabolism, glycolysis, and pyruvate metabolism were enriched in clusters 1 and 3, implying that the genes associated with these pathways were up- or down-regulated by dehydration and rapidly restored by rehydration. MAPK signaling pathway-related genes were enriched in clusters 1 and 2, whereas photosynthesis-related genes were highly enriched in cluster 3. Genes related to nitrogen and secondary metabolism were mostly enriched in clusters 3, 4, and 5. These results suggest that dehydration and rehydration treatments have variable effects on gene expression.

To confirm the expression patterns of the DEGs in the five clusters, 14 representative DEGs were selected, and their expression levels were analyzed by qRT-PCR (Figure 4). In this experiment, seedlings treated with three-day rehydration were also included to monitor the expression patterns during recovery after dehydration. In cluster 1, the genes encoding trehalose-6-phosphate synthase (LOC_Os02g54820), phosphatase (LOC_Os03g49440), and dehydrin (LOC_Os02g44870) were up-regulated by dehydration and rapidly down-regulated by rehydration (Figure 4a). In contrast, the genes encoding the hypoxia-responsive family protein (LOC_Os02g37930), MYB family transcription factor (LOC_Os04g43680), and ubiquitin family proteins (LOC_Os02g06640) were up-regulated by dehydration and slowly down-regulated by rehydration (Figure 4b). Of these, dehydration up-regulated the expression levels of LOC_Os02g37930 and LOC_Os04g43680, maintained even in the three-day rehydrated seedlings, whereas that of LOC_Os02g37930 was reduced to the expression level of the control. The DEGs in cluster 3 were down-regulated by dehydration and rapidly up-regulated by rehydration. Among these, three genes encoding aminotransferase (LOC_Os08g41990), chlorophyll A-B binding protein (LOC_Os11g13890), and oxidoreductase (LOC_Os10g35370) were validated using qRT-PCR (Figure 4c). The representative DEGs in cluster 4 included LOC_Os07g31720 (GTPase-activating protein), LOC_Os09g29540 (OsWAK82), and LOC_Os05g50340 (MYB family transcription factor). These genes showed similar expression patterns in RNA-seq and qRT-PCR (Figure 4d). Their expression levels after the three-day rehydration treatment did not fully recover to the control expression levels. The genes encoding cytochrome P450 (LOC_Os02g02230) and CHIT3 (LOC_Os04g41680) were representative DEGs in cluster 5. They were specifically up-regulated by the rehydration treatment, and their up-regulated expression levels were maintained after the three-day rehydration treatment. These results indicate that the expression patterns of selected DEGs in qRT-PCR were found to correlate with those of RNA-seq, and the genes and drought-responsive DEGs were differentially regulated during rehydration.

### 2.4. Differential Expression of Nitrogen Metabolism Related Genes during Dehydration and Rehydration

GO and KEGG enrichment analyses revealed that nitrogen metabolism-related genes were enriched in the DEG clusters. To gain insight into the biological role of nitrogen metabolism during dehydration and rehydration, we examined the expression patterns of genes involved in nitrogen uptake, assimilation, and signaling from our RNA-seq data. Of the 67 nitrogen metabolism-related genes, 31 (46.3%) were found to be differentially expressed (Figure 5a; Appendix A). These include 8 down-regulated and 17 up-regulated genes following dehydration treatment. Six genes were significantly up-regulated during rehydration treatment. Of these DEGs, nine representative genes were selected and further examined for dynamic expression pattern analysis. Among the nitrogen uptake-related genes, *OsNPF2.4* was down-regulated by dehydration and rapidly recovered by rehydration (Figure 5b). However, *OsNPF4.1* and *OsLHT1* were up-regulated by dehydration. During rehydration, the expression level of *OsNPF4.1* was decreased while that of *OsLHT1* was not significantly affected. For the nitrogen assimilation-related genes, *OsGS2* was down-regulated, whereas *OsAS1* was up-regulated by dehydration (Figure 5c). *OsGS1.2* was specifically up-regulated during rehydration. Finally, *OsMYB61*, *OsDOF18*, and *OsNLP2*, which play a role in the nitrogen signaling pathway, were differentially expressed by dehydration (Figure 5d). *OsMYB61* was down-regulated, whereas *OsDOF18* and *OsNLP2* were up-regulated by dehydration. These results imply that nitrogen metabolism-related genes are dynamically regulated in seedling shoots during dehydration and rehydration to adapt to these adverse conditions.

### 2.5. Differential Expression of Hormone-Responsive Genes and Hormone Metabolism-Related Genes during Dehydration and Rehydration

Phytohormones play an important role in the dehydration response of plants. To investigate the role of plant hormones in rice dehydration and rehydration responses, we first compared hormone-responsive rice genes with our DEG data. The hormone-responsive rice genes were obtained by reanalyzing RiceXpro microarray data [27]. As a result, 820 ABA-up-regulated genes and 747 ABA-down-regulated genes were identified from the seedling shoots treated with ABA (Appendix A). The number of genes up-regulated and down-regulated by indole acetic acid (IAA) were 269 and 112, respectively. With JA treatment, 687 and 639 genes were up-regulated and down-regulated, respectively. In the case of trans-zeatin treatment, 82 and 44 genes were identified as up-regulated and down-regulated genes, respectively. Because the number of ethylene- and gibberellin-responsive genes was limited, we excluded these genes from our analysis. By comparing the number of overlapping genes between hormone-responsive genes and DEGs identified from our data, ABA-up-regulated genes were significantly overlapped with the DEGs of Groups 1, 3, and 6, whereas ABA-down-regulated genes were highly overlapped with the DEGs of Groups 2, 4, and 5 (Figure 6a). This result is consistent with the finding that ABA is a major plant hormone involved in drought response. DEGs in Groups 1, 3, and 6 also significantly overlapped with IAA-up-regulated genes. However, the DEGs in Groups 2, 4, and 5 overlapped less with IAA-down-regulated genes. Interestingly, we found that JA-responsive genes highly overlapped with the dehydration- and rehydration-responsive genes. JA is involved in biotic stress responses in plants. However, recent studies have reported that it also plays important roles in abiotic stress responses [16]. Thus, the significant overlap between JA-responsive genes and dehydration-responsive genes might be due to crosstalk between ABA and JA [16]. Cytokinins are also known to be involved in drought response through an antagonistic relationship with ABA. However, the DEGs in our data did not significantly overlap with the trans-zeatin responsive genes relative to other hormones. This might be due to a limited number of cytokinin-responsive genes in our data analysis. A comparison between hormone-responsive genes and DEG clusters also showed similar results (Figure 6b). ABA-up-regulated, IAA-up-regulated, and JA-up-regulated genes significantly overlapped with DEGs in clusters 1, and 2, whereas ABA-down-regulated and JA-down-regulated genes highly overlapped with DEGs in clusters 4 and 5.

Because many phytohormone-responsive genes were differentially expressed during dehydration and rehydration, we further examined the expression patterns of genes involved in hormone metabolism. Of the DEGs responsive to dehydration and rehydration, those involved in hormone biosynthesis, signaling, and deactivation were selected. Hormone-responsive gene analysis revealed that several genes involved in ABA, auxin, and JA were dynamically responsive during dehydration and rehydration (Figure 6c; Appendix A). In addition, the expression patterns of some genes involved in the metabolism of other hormones were also regulated by dehydration and rehydration. These results imply that crosstalk between ABA, JA, and other hormones might play a role in dehydration and rehydration responses in rice.

### 2.6. Differential Expression of Transcription Factor Genes during Dehydration and Rehydration

Many transcription factors play crucial roles in responses to drought stress by regulating the expression of other genes [3]. In addition, their expression has been shown to be regulated by drought stress treatment. To investigate the responses of rice transcription factors during dehydration and rehydration, we examined the expression patterns of 2048 transcription factors registered in the Rice Transcription Factor Phylogenomics Database [28]. We found that 445 (21.7%) transcription factors were significantly regulated by dehydration and rehydration treatments (Figure 7a; Appendix A). It is worth noting that more transcription factors encoding DEGs were identified in Groups 1 and 3 than in Groups 2 and 4. These ratios were higher than those from all DEGs shown in Figure 1c. This result indicates that there are more drought-up-regulated transcription factors than drought-down-regulated ones (Figure 7b).

The DEGs belonging to the AP2/EREBP, NAC, MYB, WRKY, bZIP, and bHLH transcription factor families were over-represented in Groups 1, 3, and 6 (Figure 7c). This means that the DEGs in these families were mostly up-regulated rather than down-regulated by the dehydration treatment. Interestingly, the DEGs of MYB and Homeobox families were identified not only in Groups 1, 3, and 6 but also in Groups 2, 4, and 5. When the expression patterns of DEGs of four major transcription factor families were examined in detail, most of the DEGs in the AP2/EREBP and NAC families were up-regulated by dehydration, and their expression levels were recovered by rehydration (Figure 7d). In contrast, in the case of MYB transcription factors, 18 genes were up-regulated and 17 were down-regulated by dehydration treatment. In the case of the WRKY family, 17 and 4 genes were up-regulated and down-regulated, respectively, by dehydration. Finally, it is noteworthy that five genes encoding WRKY transcription factors were significantly up-regulated by rehydration treatment. The dynamic expression patterns of these transcription factors may contribute to the regulation of downstream gene expression to adapt to dehydration and rehydration stresses.

## 3. Discussion

In this study, we examined the genome-wide gene expression patterns in rice seedling shoots during dehydration and rehydration. As a result, approximately 14% of the rice genes were differentially expressed, and drought-responsive genes were further classified as those whose expression was rapidly or slowly restored after rehydration. In particular, genes involved in photosynthesis, nitrogen metabolism, hormone metabolism and signaling, and transcriptional regulation dynamically respond to dehydration and rehydration.

Photosynthesis-related genes are rapidly down-regulated by dehydration [29]. Our RNA-seq data revealed that the expression levels of these genes rapidly recovered by dehydration. This finding was supported by the photosystem II efficiency measured by Fv/Fm values. Fv/Fm values rapidly decreased with dehydration and recovered to control levels in healthy parts of the leaves within a day. Because of decreased photosynthetic efficiency, carbon metabolism-related genes were also affected by drought stress. This suggests that genes related to nitrogen metabolism may also be regulated by drought stress. Indeed, it has been reported that nitrogen metabolism-related genes are dynamically regulated by drought stress in Arabidopsis, soybean, apples, and maize [30,31,32,33]. Our RNA-seq data analysis supported that some rice nitrogen metabolism-related genes are significantly regulated by dehydration and rehydration. For instance, *OsNPF2.4*, *OsGS2*, and *OsMYB61* are down-regulated by dehydration and up-regulated by rehydration. In contrast, *OsNPF4.1/SP1*, *OsAS1*, and *OsDOF18* showed opposite expression patterns, which were up-regulated by dehydration and down-regulated by rehydration. These results imply that differential expression of nitrogen metabolism is required to balance carbon and nitrogen metabolism during adaptation to drought stress. Further studies are required to understand the regulation of these genes in specific cell types during dehydration and rehydration.

ABA is a major phytohormone involved in drought stress response. Our data support the important role of ABA in the positive correlation between drought- and ABA-responsive genes. ABA-inducible genes tended to be up-regulated by dehydration, whereas ABA-repressible genes tended to be down-regulated by dehydration. Given that other phytohormones, including cytokinin, auxin, brassinosteroid, ethylene, gibberellin, and JA, crosstalk with ABA during drought stress responses, it is assumed that genes regulated by other phytohormones are also differentially expressed during dehydration and rehydration. We only found that auxin- and JA-responsive genes overlapped significantly with our DEGs. This might be due to the limited numbers of cytokinin-, brassinosteroid-, ethylene-, and gibberellin-responsive DEGs in our analysis. Nevertheless, several genes involved in the biosynthesis and signaling of cytokinin, ethylene, and gibberellin were found to be differentially expressed during dehydration and rehydration. These results imply that the extent of the regulated gene numbers may vary depending on the crosstalk between ABA and other phytohormones. We noticed that JA-responsive genes significantly overlapped with drought-responsive genes compared to other phytohormone-responsive genes. JA-induced genes overlapped significantly with drought-induced genes, whereas JA-repressible genes overlapped with drought-repressible genes. Several studies have shown that JA is involved in drought responses via the interaction between JA and ABA signaling [34,35,36,37,38,39]. In rice, *OsJAZ1*, a JA co-receptor and transcriptional repressor, plays a negative role in drought resistance [17]. We found that *OsJAZ1*, *OsJAZ2*, *OsJAZ5*, *OsJAZ6*, *OsJZA7*, *OsJAZ9*, *OsZIM8*, and *OsZIM18* were induced by dehydration, and their expression was differentially regulated during rehydration (Figure 5c). Moreover, dehydration down-regulated the expression of OsJMT2, OsJMT3, and JA biosynthetic genes. These results suggest that other JAZ proteins and JA signaling may also play a role in drought stress responses.

Dynamic gene expression changes during dehydration are attributed to drought-responsive TFs. Our RNA-seq data analysis revealed that ~22% of the rice TFs were differentially expressed during dehydration and rehydration. These include well-known drought stress-related TF families such as AP2/EREBP, MYB, WRKY, bZIP, and bHLH. Notably, the number of drought-inducible TF genes was much greater than that of the drought-repressible TF genes. Indeed, several studies have reported that the overexpression of drought-inducible TFs results in increased drought resistance. However, functional studies on drought-repressible TFs are relatively limited. As the knockout of drought-repressible TFs may also be useful for developing drought-resistant plants, the list of rice drought-repressible genes in our data will be a valuable resource for further study. In addition, it would be intriguing to study how drought-repressible genes are regulated by transcriptional and post-transcriptional regulation, which cannot be explained by the small number of drought-repressible TF genes.

There are several limitations in interpreting our results to understand dehydration- and rehydration-responsive gene expression. This is because the rehydrated samples were mixed tissues that were recovered and damaged. Indeed, healthy parts of leaves after rehydration showed recovered photosynthetic efficiency, while damaged parts of leaves from the same plant showed even lower photosynthetic efficiency compared to dehydrated leaves. Thus, gene expression patterns during rehydration represent the combined expression levels in both tissues. In addition, since bulk RNA-seq data reflect the overall status of gene expression in various cell types, it is difficult to distinguish tissue- or cell-type specifically expressed genes. Further studies using single-cell RNA-seq technology may reveal high-resolution gene expression patterns during dehydration and rehydration. Another limitation is that our data were obtained from the 8 h dehydration treatment, which is an acute condition and may not properly reflect the drought stress under real field conditions. Nevertheless, we expect that our data will be a useful resource for understanding drought stress responses in rice and will provide a useful gene list to develop drought-resistant crop plants.

## 4. Materials and Methods

### 4.1. Plant Growth Conditions and Stress Treatment

Dehusked rice seeds (*Oryza sativa* L. ssp. Japonica cv. Nipponbare) were sterilized and germinated on half-strength Murashige and Skoog (MS) medium (Duchefa Biochemie, Haarlem, Netherlands) containing 0.2% (*w*/*v*) phytagel (Sigma-Aldrich, St. Louis, MO, USA) in a sterile plastic box. The plants were grown in a growth chamber under continuous light conditions at 25 °C. For the dehydration treatment, 10-day-old seedlings were air-dried with a paper towel for 8 h as described in our previous studies [40,41]. Control plants were sampled at the same time as the dehydration treatment. Rehydration was performed by re-watering the 8 h dehydrated seedlings for 24 or 72 h. Biological replicates of seedling shoots from each treatment were collected, immediately frozen in liquid nitrogen, and stored at −80 °C until RNA extraction. Seedling shoots were collected by cutting above 0.5 Cm from the seeds. Each biological replicate was from a pool of 20 individual plants.

### 4.2. Measurements of Chlorophyll Fluorescence

The chlorophyll fluorescence (Fv/Fm) was measured using a portable fluorometer (FluorPen, FP1000; Photon Systems Instruments, Drásov, Czech Republic). The Fv/Fm ratio was measured in the fully expanded third or fourth leaf of each plant after 20 min of dark adaptation. For rehydrated seedlings, the Fv/Fm ratios of the weak and healthy parts were separately measured.

### 4.3. RNA-Seq Library Construction and Sequencing

Total RNA was isolated from the tissues using the Tri-RNA Reagent (Favorgen, FATRR 001; Ping Tung, Taiwan). The RNA concentration was measured using a NanoDrop 2000 spectrophotometer (Thermo Fisher Scientific, Waltham, MA, USA; ND-2000). The RNA quality was checked using a Bioanalyzer (Agilent Technologies, Santa Clara, CA, USA; 2100 Bioanalyzer) and confirmed more than 8 RNA integrity numbers (RIN). RNA-seq libraries were constructed from two biological replicates using a TruSeq Stranded mRNA Library Prep Kit (Illumina, San Diego, CA, USA; RS-122-2101), according to the manufacturer’s protocol. RNA-seq libraries were constructed using the indexed adaptors provided in the kit and pooled for the sequencing. RNA-seq libraries were sequenced with a paired 2 × 75 bp length on the Illumina MiSeq platform.

### 4.4. Bioinformatic Analyses of RNA-Seq Data

The paired-end raw sequencing reads were cleaned by pre-processing with FastQC (Q20 and Q30 reads ≥ 80%) and adaptor trimming in a MiSeq platform. Mapping to the rice genome (Rice Genome Annotation Project, MSU v7.0) was conducted using the clean reads with 1 nt mismatch allowance using the CLC Genomics Workbench 20.0.4. Gene expression levels were normalized to reads per kilobase of transcript per million mapped reads (RPKM) using the same program. Differentially expressed genes were identified using an EdgeR program and defined as fold changes > 2 and adjusted *p*-values < 0.01. The *p*-values were adjusted using Benjamini and Hochberg’s approach for controlling the false discovery rate (FDR). Gene ontology (GO) enrichment analysis was conducted using agriGO version 2.0 using the DEGs in each group [42]. The Kyoto Encyclopedia of Genes and Genomes (KEGG) pathway was analyzed using KOBAS [43]. GO and KEGG terms with a *p*-values < 0.01 was considered as significantly enriched. Principal component analysis (PCA) was performed using RPKM values of all the mapped genes and visualized using TBtools [44]. DEG clustering was conducted using K-means with the Multiple Array Viewer (MeV) v4.9.0 [45]. The input data for DEG clustering were normalized z-scores. The distance metric was set as Pearson correlation, and k-mean values were set as 10. Manual curation was performed to select the five main clusters. To retrieve hormone-responsive genes, publicly available microarray data were downloaded from RiceXPro [27]. The hormone-responsive genes were identified with the criterial: fold changes > 2 and *t*-test *p*-values < 0.01.

### 4.5. Quantitiative RT-PCR

First-strand cDNAs were synthesized using a ReverTra Ace qPCR RT Master Mix with a gDNA Remover kit (Toyobo, FSQ-301; Osaka, Japan). qRT-PCR was performed using a KOD SYBR qPCR Mix kit (Toyobo, QKD-201; Osaka, Japan) with an Exicycler 96 Real-Time Quantitative Thermal Block System (Bioneer, Daejeon, Republic of Korea). The expression levels were calculated from two technical replicates of two biological replicates (n = 4) using the 2^−ΔCT^ method [46]. The rice *Actin1* gene was used as an internal control for normalization. The representative DEGs were selected based on the following criteria: being among the top 100 significant *p*-values, having an expression level of more than 10 RPKM in the highest expression condition, and having a functional annotation. Primers used in this study are listed in Appendix A.

### 4.6. Statistical Analyses

Statistical significance of multiple sample comparison was determined using ANOVA with Tukey’s honest significant difference (HSD) test using R software (version 4.3.0). The correlation R^2^ values were calculated by a Pearson correlation method using R software (version 4.3.0). The statistical significance of the overlap between two gene groups was obtained from the website: http://nemates.org/MA/progs/overlap_stats.html (accessed on 7 May 2023).

## 5. Conclusions

In this study, we examined the genome-wide gene expression patterns in rice during dehydration and rehydration. The results showed that approximately 14% of the rice genes were differentially expressed, and drought-responsive genes were dynamically regulated during rehydration. Those DEGs were either rapidly or slowly recovered to control levels. Furthermore, some DEGs were preferentially regulated during rehydration, indicating that the rehydration response does not necessarily equate to a return to the control state. Our data showed the potential role of nitrogen metabolism and JA signaling during the drought stress response. These observations suggest that the regulation of the carbon–nitrogen balance and crosstalk between abiotic and biotic stress responses are significant components of the drought stress response. The transcriptome data in this study could be a useful resource for understanding drought stress responses in rice. We believe that our data are providing a valuable gene list for developing drought-resistant crop plants.

## Figures and Tables

**Figure 1 ijms-24-08439-f001:**
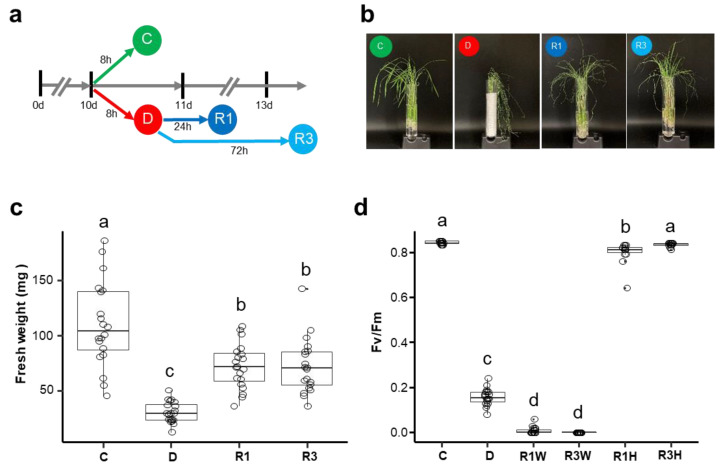
Phenotypes of rice seedlings under dehydration and rehydration stress conditions. (**a**) Experimental scheme showing the time course of stress treatment. Ten-day-old seedlings were exposed to 8 h dehydration (D) or control (C) conditions. Rehydration was treated for one day (R1) or three days (R3) using 8 h dehydrated seedlings. (**b**) Phenotypes of control (C), dehydration-treated (D), and one day- (R1) or three day- (R3) rehydrated plants. (**c**) Fresh weights of rice seedlings under control (C), dehydration (D), and rehydration (R1 and R3) conditions. Fresh weights were measured using 20 seedlings at the indicated time points. (**d**) Chlorophyll fluorescence (Fv/Fm) of rice seedlings under control (C), dehydration (D), and rehydration (R1W, R1H, R3W, and R3H) conditions. For the rehydrated plants, Fv/Fm values of weak (R1W and R3W) and healthy (R1H and R3H) leaves were measured separately. The number of seedlings used for the Fv/Fm measurement was 20 except for R1H (n = 15) and R3H (n = 16). For the box plots in c and d, the boxes represent the interquartile range (IQR) showing the lower (Q1) and upper (Q3) quartiles surrounding the median (central thick line), and whiskers represent the minimum (Q1 − 1.5 × IQR) and maximum (Q3 + 1.5 × IQR) values. Individual values are indicated as circles. Significant differences were determined by ANOVA with Tukey’s HSD and indicated by different letters.

**Figure 2 ijms-24-08439-f002:**
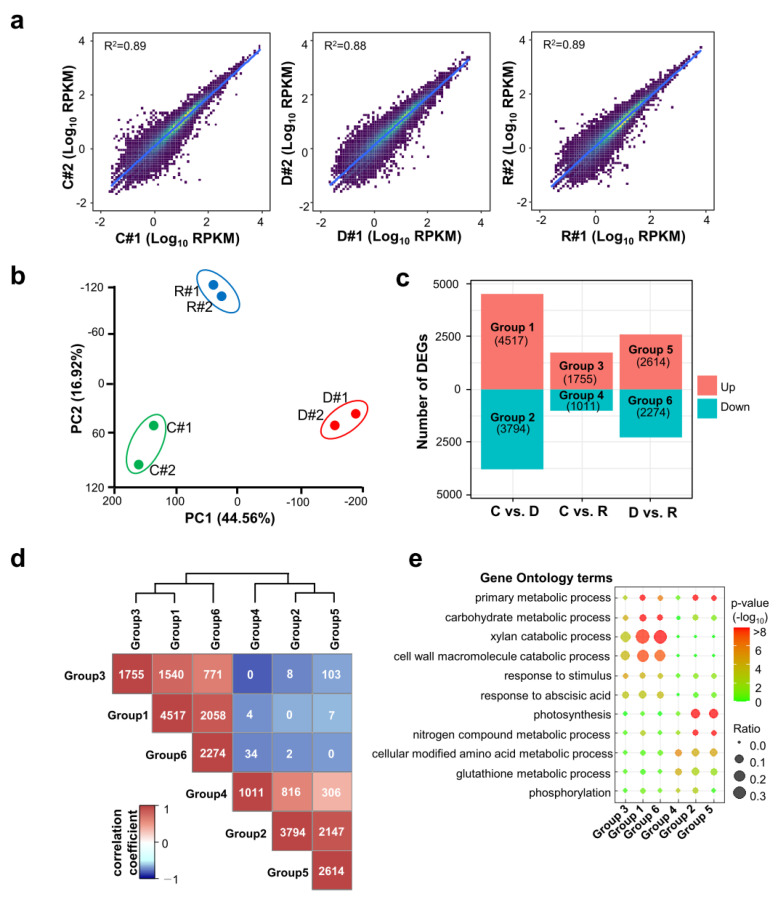
Identification of differentially expressed genes by dehydration and rehydration stress treatments. (**a**) Density scatter plots showing the correlations of gene expression levels (Log_10_RPKM) between biological replicates (#1 and #2). C: control, D: dehydration-treated, R: rehydration-treated seedlings. Dot colors represent density of gene numbers. R^2^ indicates the Pearson’s correlation coefficient. (**b**) Principal component analysis of normalized RNA-seq data. Percentages represent variance captured by principal components 1 and 2 in each analysis. The colored circles in the plot represent individual libraries. Two biological replicates of RNA-seq libraries from control, dehydration, and rehydration are grouped in green, red, and blue oval-shaped rings, respectively. (**c**) Differentially expressed genes (DEGs) categorized into six groups by up- or down-regulated genes between different conditions. The number of DEGs in each group is shown in the parentheses. (**d**) Correlation matrix of pairwise DEG group comparison. Heatmap shows the numbers of overlapping DEGs and the Pearson’s correlation coefficient between the groups. The number of overlapping DEGs is shown in the boxes. (**e**) Dot plot showing enrichment of gene ontology biological processes for the DEGs in six DEG groups. Colors indicate the *p*-values from Fisher’s exact test, and dot sizes are proportional to the number of DEGs in the given GO terms.

**Figure 3 ijms-24-08439-f003:**
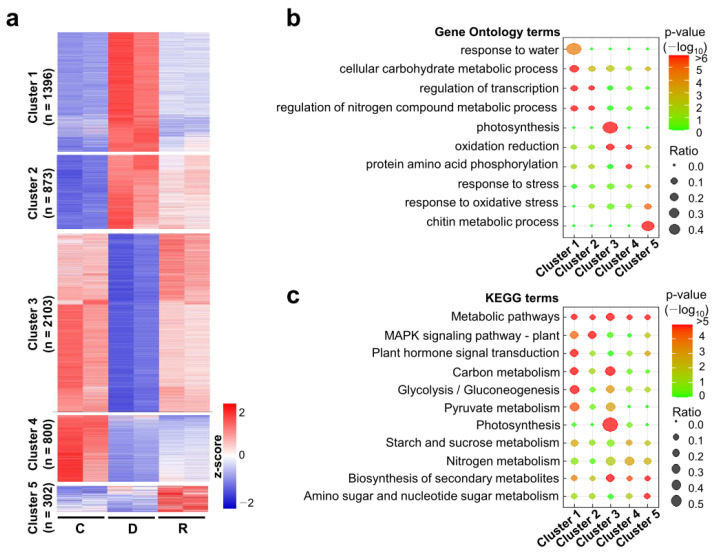
Clustering, gene ontology, and KEGG analysis of the differentially expressed genes. (**a**) Heatmap showing the five main clusters identified by a hierarchical clustering of DEGs. The expression level of each DEG is represented as normalized z-score. (**b**) Gene ontology (GO) analysis of DEGs in five clusters. *Y*-axis indicates the enriched GO terms in the biological process. Dot colors and size represent *p*-values from Fisher’s exact test proportional to the number of DEGs in the given GO terms. (**c**) KEGG pathway analysis of DEGs in five clusters. *Y*-axis indicates the enriched KEGG pathways. Dot colors and size represent *p*-values from Fisher’s exact test proportional to the number of DEGs in the given GO terms.

**Figure 4 ijms-24-08439-f004:**
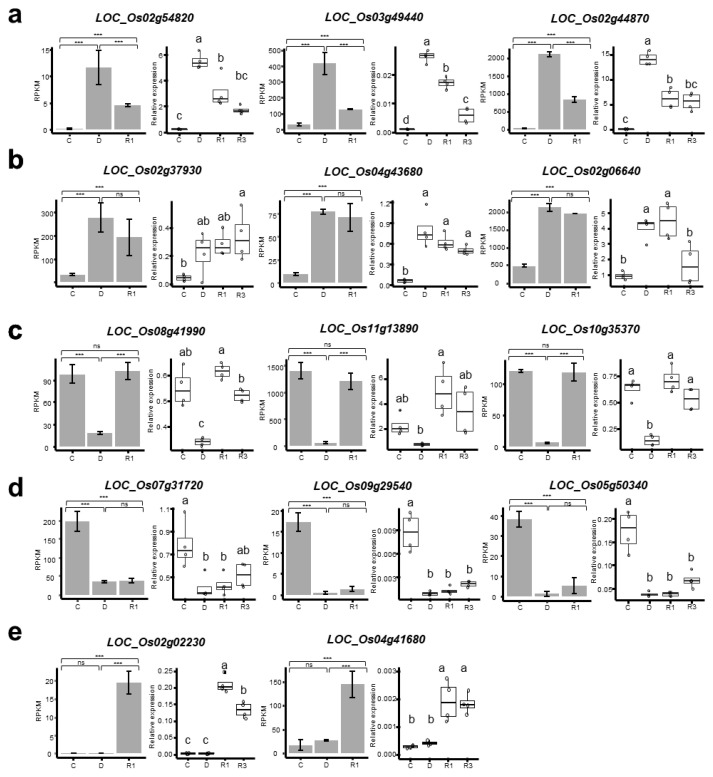
Validation of the differentially expressed genes in the five main clusters. (**a**–**e**) Two to three representative genes from each cluster are selected for showing the RNA-seq expression levels and qRT-PCR validation. RNA-seq data are shown as bar plots with reads per kilobase of transcript, per million mapped reads (RPKM) values of two biological replicates (n = 2). Statistical significance was determined by EdgeR program. The asterisks indicate significant difference (*** *p* < 0.001; and ns: *p* > 0.01). Relative expression levels of each gene were calculated from two technical replicates of two biological replicates (n = 4) of control seedlings (C), dehydration-treated seedlings (D), and rehydration-treated seedlings for 1 d (R1) and 3 d (R3) by qRT-PCR and shown as box plots. OsAct1 was used as internal control. Significant differences were determined by ANOVA with Tukey’s HSD test and indicated by different letters. The genes and their encoding proteins are as follows: *LOC_Os02g54820*; trehalose-6-phosphate synthase, *LOC_Os03g49440*; phosphatase, *LOC_Os02g44870*; dehydrin, *LOC_Os02g37930*; hypoxia-responsive family protein, *LOC_Os04g43680*; MYB family transcription factor, *LOC_Os02g06640*; ubiquitin family protein, *LOC_Os08g41990*; aminotransferase, *LOC_Os11g13890*; chlorophyll A-B binding protein, *LOC_Os10g35370*; oxidoreductase, *LOC_Os07g31720*; GTPase activating protein, *LOC_Os09g29540*; OsWAK82, *LOC_Os05g50340*; MYB family transcription factor, *LOC_Os02g02230*; cytochrome P450, *LOC_Os04g41680*; CHIT3.

**Figure 5 ijms-24-08439-f005:**
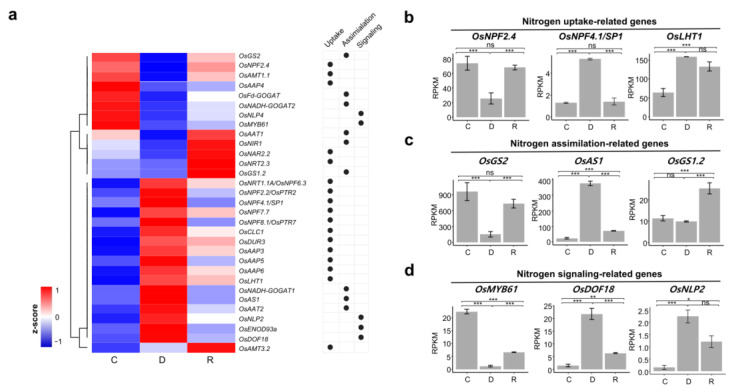
Gene expression profile of nitrogen metabolism-related genes under dehydration and rehydration stress conditions. (**a**) Heatmap showing the hierarchical clustering of the DEGs involved in nitrogen metabolism. The expression levels of each gene are normalized into z-score. Dots on the right side represent the role of each gene in nitrogen metabolism pathways. C: control, D: dehydration, R: rehydration. (**b**–**d**) Expression patterns of selected genes involved in nitrogen uptake (**b**), nitrogen assimilation (**c**), and nitrogen signaling pathway (**d**). The normalized expression levels of each gene under control (C), dehydration (D), and rehydration (R) conditions are represented from RNA-seq data. Statistical significance was determined by EdgeR program. The asterisks indicate significant difference (*** *p* < 0.001; ** *p* < 0.05; * *p* < 0.01; and ns: *p* > 0.01).

**Figure 6 ijms-24-08439-f006:**
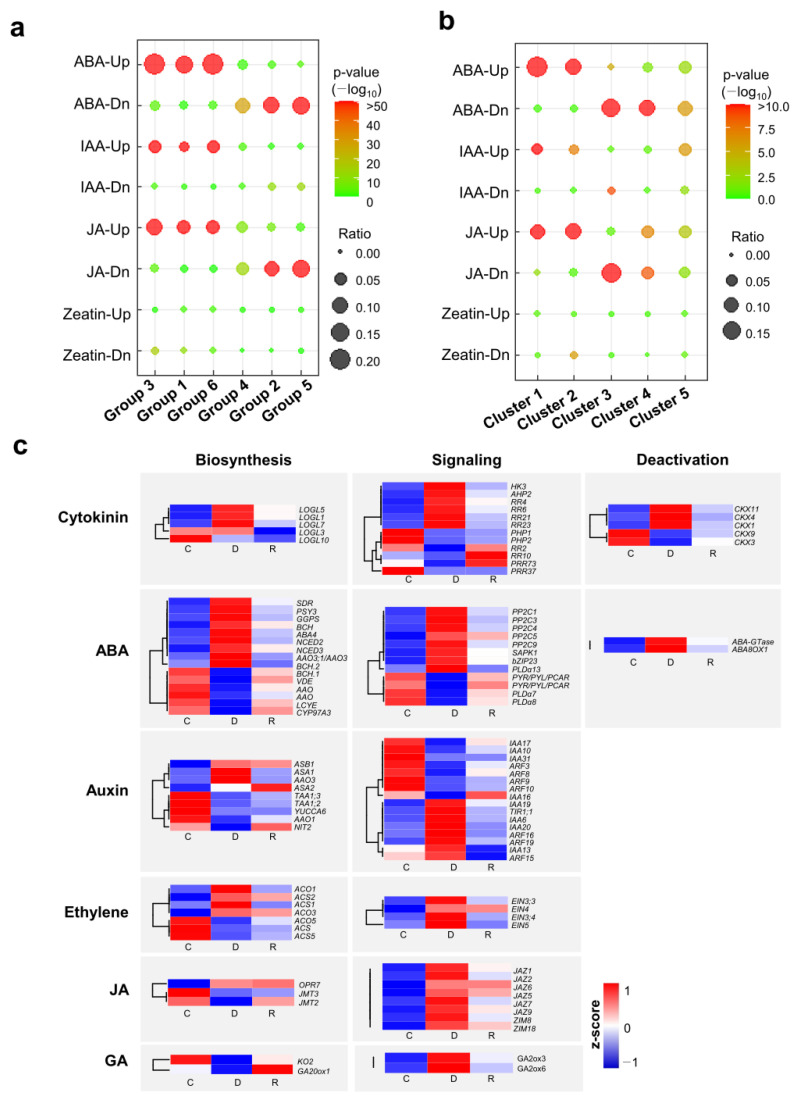
Gene expression profile of plant hormone-related genes under dehydration and rehydration stress conditions. (**a**) Correlation of plant hormone-responsive gene sets and dehydration- and rehydration-responsive DEG groups. Dot colors represent Pearson’s correlation *p*-values and dot sizes are proportional to the number of overlapping DEGs. (**b**) Correlation of plant hormone-responsive genes and five main DEG clusters. Dot colors represent Pearson’s correlation *p*-values and dot sizes are proportional to the number of overlapping DEGs. (**c**) Heatmap showing the expression patterns of DEGs related with plant hormone biosynthesis, signaling, and deactivation. The expression levels are normalized to z-score. ABA: abscisic acid, IAA: indole 3-acetic acid, JA: jasmonic acid, GA: gibberellic acid.

**Figure 7 ijms-24-08439-f007:**
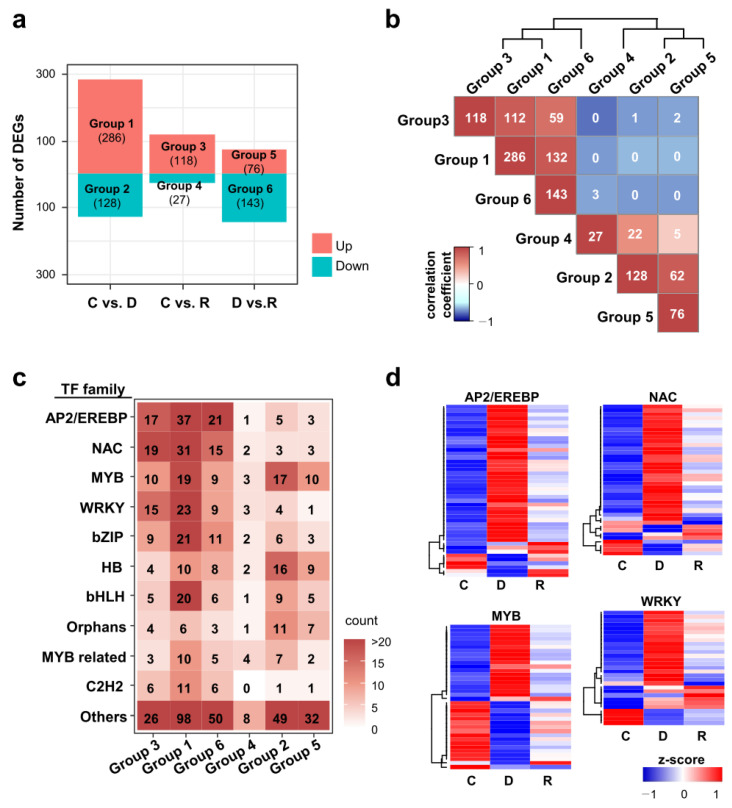
Differentially expressed transcription factor genes under dehydration and rehydration stress conditions. (**a**) Differentially expressed transcription factor (TF) genes categorized into six groups by up- or down-regulated genes between different conditions. The number of DEGs in each group is shown in the parentheses. (**b**) Correlation matrix of pairwise TF DEG group comparison. Heatmap showing the numbers of overlapping TF DEGs and the Pearson’s correlation coefficient between the groups. (**c**) Heatmap showing the gene numbers of TF families in each DEG group. Gene counts are shown in colors. (**d**) Heatmap showing the hierarchical clustering of TF DEGs in AP2-EREBP, NAC, MYB, and WRKY families. The expression levels are normalized to z-score.

## Data Availability

The raw reads of RNA-seq were deposited at the Sequence Reads Archive of the National Center for Biotechnology Information under accession number GSE222438.

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
