# Peer review of "Comprehensive Analysis of Rice Seedling Transcriptome during Dehydration and Rehydration"

_ijms, 2023, doi:10.3390/ijms24098439_

Round 1
Reviewer 1 Report
The topic of the present paper „Comprehensive Analysis of Rice Seedling Transcriptome During Dehydration and Rehydration” is very interesting for reader, drought being a harmful abiotic stress that threatens the growth, development, and yield of rice plants.
In the present manuscript, the authors examined the dynamic gene expression patterns in rice seedling shoots during dehydration and rehydration using RNA-seq analysis.
The authors data showed the potential role of nitrogen metabolism and jasmonic acid signaling during the drought stress response. So, they conclude that the transcriptome data in this study could be a useful resource for understanding drought stress responses in rice and are providing a valuable gene list for developing drought-resistant crop plants.
Finally, I conclude that:
- - the topic of the present manuscript is relevant on the field;
- - the introduction provides sufficient background and includes relevant references;
- - the design research is well explained, so I consider that the authors should not consider any improvements;
- - the results of this study provide useful information for the development of drought-tolerant rice crops;
- -the conclusions are consistent because evidence the presented arguments;
- - the reference list is variously and recently;
- - the manuscript is well written, and the text is easy to read.
Author Response
Reviewer's comments:
The topic of the present paper „Comprehensive Analysis of Rice Seedling Transcriptome During Dehydration and Rehydration” is very interesting for reader, drought being a harmful abiotic stress that threatens the growth, development, and yield of rice plants.
In the present manuscript, the authors examined the dynamic gene expression patterns in rice seedling shoots during dehydration and rehydration using RNA-seq analysis.
The authors data showed the potential role of nitrogen metabolism and jasmonic acid signaling during the drought stress response. So, they conclude that the transcriptome data in this study could be a useful resource for understanding drought stress responses in rice and are providing a valuable gene list for developing drought-resistant crop plants.
Finally, I conclude that:
- the topic of the present manuscript is relevant on the field;
- the introduction provides sufficient background and includes relevant references;
- the design research is well explained, so I consider that the authors should not consider any improvements;
- the results of this study provide useful information for the development of drought-tolerant rice crops;
-the conclusions are consistent because evidence the presented arguments;
- the reference list is variously and recently;
- the manuscript is well written, and the text is easy to read.
Answer: We appreciate your valuable comments and hope that our manuscript will be a useful resource for the plant science community.
Reviewer 2 Report
Park and Jeong submitted a manuscript titled "Comprehensive Analysis of Rice Seedling Transcriptome During Dehydration and Rehydration" for publication in IJMS.
Though the work is interesting, it is very commonplace to study drought affecting plant physiology through omics. However, the analyses and conclusions were done well, and well written. The ms can be accepted for publication in the present form.
Author Response
Reviewer's comments:
Park and Jeong submitted a manuscript titled "Comprehensive Analysis of Rice Seedling Transcriptome During Dehydration and Rehydration" for publication in IJMS.
Though the work is interesting, it is very commonplace to study drought affecting plant physiology through omics. However, the analyses and conclusions were done well, and well written. The ms can be accepted for publication in the present form.
Answer: We appreciate your valuable comments and hope that our manuscript will be a useful resource for the plant science community.
Reviewer 3 Report
It appears that dehydration for 8 hours and drought are used interchangeably in the manuscript. I don't think drought response and dehydration for 8 hours response are similar. It would be appropriate to use acute dehydration or dehydration for 8 hours in the manuscript rather than drought. Findings based on acute dehydration can be useful to develop strategies for drought tolerance, but can not be directly extrapolated to drought mechanisms in rice.
It's easy to follow, however, minor editing of the English language is required.
Author Response
Reviewer's comments:
It appears that dehydration for 8 hours and drought are used interchangeably in the manuscript. I don't think drought response and dehydration for 8 hours response are similar. It would be appropriate to use acute dehydration or dehydration for 8 hours in the manuscript rather than drought. Findings based on acute dehydration can be useful to develop strategies for drought tolerance, but can not be directly extrapolated to drought mechanisms in rice.
Answer: We agree with your concern about the 8-h dehydration treatment. Thus, we have changed the ‘drought’ to ‘dehydration’ in several sentences of the manuscript. In addition, we have indicated the limitation of our study in line 479-281 as follows:
“Another limitation is that our data were obtained from the 8-hour dehydration treatment, which is an acute condition and may not properly reflect the drought stress under real field conditions.”
It's easy to follow, however, minor editing of the English language is required.
Answer: We apologize for any inconvenience caused by the errors in our manuscript. Our original manuscript has been edited by the professional English editing service, Editage. We will make further edits upon request during the final proof stage.